# Impact of Food Sustainability Labels on the Price of Rice in Online Sales

**DOI:** 10.3390/foods11233781

**Published:** 2022-11-23

**Authors:** Xinyu Ma, Ziqi Liu, Ting Meng, Wojciech J. Florkowski, Yueying Mu

**Affiliations:** 1College of Economics and Management, China Agricultural University, Beijing 100083, China; 2Academy of Food Policy and Economics (AGFEP), Beijing Food Safety Policy and Strategy Research Base, China Agricultural University, Beijing 100083, China; 3Department of Agricultural Economics, University of Georgia, Griffin, GA 30223-1797, USA

**Keywords:** green food, price premium, hedonic price model, e-commerce platforms

## Abstract

Currently, the quality and safety of agricultural products and the enhancement of the agroecological environment are widely discussed. In response to solving the problem of insufficient exploitation of the market potential regarding sustainable agricultural products, this study uses rice on e-commerce platforms as an empirical case and constructs a hedonic price model aiming to explore the impact of the sustainable label on the price premium of agricultural products. The results show that: (1) There is a significant price premium for rice with sustainable labeling over ordinary rice, which is about 47.55%. In addition, within the types of sustainable labels, the price premium for rice with an organic food label is higher than that of rice with a green food label. (2) Except for the sustainable label, factors affecting the price premium of rice products include e-commerce platforms, rice varieties, package types, and whether it is imported. The price premium indicates the actual recognition and preference of consumers for agricultural products with the sustainable label. Departments of agricultural and food management departments should cooperate to improve the agricultural certification system (i.e., the sustainable label), further unblock a positive market mechanism of “green label—high quality—good price”, and facilitate the green transformation of China’s agricultural production from the consumer side.

## 1. Introduction

Driven by emerging environmentalism and health concerns about exposure to pesticides and antibiotics, organic agriculture received much attention in the 1980s, leading to the formation of eco-label systems in the food sector in Western countries [1]. Eco-label, also known as a sustainable label, has been an effective tool in affecting consumers’ decision making concerning the purchase of sustainable products [2,3]. Theoretically, it is also an incentive for producers to increase the environmental standards of products [2,4]. However, such results are not always true in practice, and some empirical findings indicate that the price premium at the retail level does not necessarily imply the premium at the producer level [5,6].

In China, the sustainable label system in the food sector was established in the 1990s [7]. Sustainable labels in this paper refer to “Green Food” labels and “Organic food” labels (Table 1). In 1996, the China Green Food Development Center promulgated the “Green Food” certification, which is unique in China. The “Green Food” certification can be divided into two different levels: A-class (allowing the use of a certain amount of chemicals) and AA-class (using internationally accepted norms for organic food laid down by IFOAM, equivalent to “organic food”) [8]. Due to the intensification of the “organic food” certification, China Green Food Development Center officially suspended the certification of AA-class “green food” in June 2008 [9]. There are specific technical standards for different agricultural products, such as Environmental Quality of Production (NY/T 391), General Guidelines for Packaging (NY/T 658), Guidelines for Pesticide Use (NY/T 393), and Guidelines for Fertilizer Use (NY/T 394).

China’s economy has grown steadily, leading to a significant increase in consumer income. Due to the improved living standards and numerous food crises (e.g., Sanlu milk powder scandal, cadmium rice incident, and lean meat essence incident), Chinese consumers are increasingly concerned about the safety and quality of food products [11]. In 2018, the market size of organic food consumption in China exceeded CNY 62.4 billion (the exchange rate of USD to CNY was 6.62 on average in 2018), making it the third-largest consumer of organic food in the world [12]. Additionally, previous “high input, high output” methods in agriculture caused serious pollution, which not only restricted the development of agriculture but also threatened China’s food security [8]. Thus, the need to increase the production and manufacturing of sustainable and high-quality agricultural products and to promote the green development of agriculture has become more urgent.

In general, the price of green agricultural products (products with a “Green Food” label or “Organic Food” label) is higher than that of conventional agricultural products due to the costs associated with obtaining and using the sustainable label [13]. The majority of consumers are willing to pay a price premium for sustainable agricultural products [14,15]. However, in reality, sustainable labeling does not always have the intended results [3].

China’s agricultural products market has developed rapidly in recent years. The gross output value of China’s agriculture was CNY 7.17 trillion (the exchange rate of USD to CNY was 6.90 on average in 2020), and online sales of agricultural products accounted for 5.8% (CNY 415.89 billion), an increase of 26.2% year-on-year [16]. With the popularity of online shopping, people are increasingly inclined to purchase daily food through e-commerce platforms [12]. Compared with the traditional retail pattern, e-commerce platforms have the advantage of cost reduction and demand enhancement [17]. For green agricultural products, the target group is more often young and middle-aged people in urban areas [11]. E-commerce platforms, as an effective tool to address issues related to product circulation and marketing, are well suited for the sale of green agricultural products [18]. However, research on consumer behavior related to sustainable agricultural products on fresh food e-commerce platforms is minimal [12,19].

Therefore, the objective of this study is to examine the price premium of sustainable labeling on agricultural products on e-commerce platforms based on the revealed preference theory. This study contributes to the current literature in the following aspects. First, it investigates the food price premium of sustainable labeling on the basis of the actual transaction data. As compared with stated preferences data, actual price premium can avoid a hypothetical bias from the willingness to pay data to reveal consumer choices better. Second, this study explores the price premium of sustainable labeling on e-commerce platforms. Results from the emerging e-commerce platforms enrich the current studies focused on the traditional offline market. Because of increasing concerns for the environment, sustainable-labeled agri-food products have been developed in many other countries worldwide. Results based on actual rice price data from e-commerce platforms in China provide broader insights for international audiences.

This study is organized into six sections. Section 2 introduces relevant studies that explore how sustainable labeling affects the food market through consumer willingness to pay and market price. Section 3 presents a hedonic pricing model to explain the attributes of food prices. Section 4 summarizes the data source and descriptive statistics. Section 5 discusses the estimation results of the price premium from the empirical analysis. Finally, Section 6 presents the conclusions of the study, the limitations of the data and results, and possible implications for policy design and implementation.

## 2. Relevant Studies

Price premium refers to the excess prices paid over and above the “fair” price that is justified by the “true” value of the product [20,21]. As for the price premium related to sustainable labels, it is defined as an amount of money that buyers were keen to disburse to protect the environment [22,23].

Numerous studies have estimated the price premium that consumers are willing to pay for sustainable labels in the food sector [10]. Methods for measuring the price premium fall into two categories, corresponding to the theory of revealed preference and stated preference [24]. The former includes auctions, laboratory experiments, field experiments, and market data, and the latter includes discrete choice analysis [25], conjoint analysis, customer surveys [26], and expert judgments. Evidence from stated preference surveys indicates that consumers generally express a preference for products with sustainable labels [11,26]. People are willing to pay USD 21.95 extra per year for organic CAS milk, according to a survey using the contingent valuation method (CVM) in Taiwan [27]. Through a meta-analysis of 80 studies worldwide, the research focused on a broad area of sustainable food products suggests that the overall WTP premium for sustainability (in percentage terms) is 29.5% on average [28]. Its results also indicate that the WTP estimate conducted by the hypothetical approach (choice experiment and contingent valuation method) is higher than the non-hypothetical one due to hypothetical bias. That is to say, there is a research gap between the WTP and the actual purchasing behavior, which can bias producers’ decisions [29].

Those relying on actual market data (a revealed preference method) are limited, yet confirm the existence of price premiums in the retail market for sustainable labels [30,31]. Historically, hedonic analysis is widely used for scanner data or privately collected secondary data when estimating implicit prices in the food sector [32]. Several scholars used it for measuring the price premium of differentiated food (wine, egg, olive oil) product attributes [33,34,35,36]. By comparing the price premiums for sustainability attributes in Chinese online and offline markets, Jiang et al. [19] find that the “Green Food” label could gain a price premium in the online market but not in the offline market.

A number of studies examined the motivations of consumers’ price premiums for sustainable labels [37,38]. Using structural equation modeling, Voon et al. [39] found that attitude and subjective norms exerted significant positive effects on willingness to pay, which positively affects actual purchase. Lin et al. [12], based on a survey of consumers who bought organic foods online, concluded that product characteristics and platform characteristics significantly impact the perceived utilitarian value and perceived hedonic value for consumers, and perceived value plays a critical mediating role in influencing product characteristics and platform characteristics on consumers’ continuous purchase intention.

In addition to the motivations discussed above, much research focuses on the factors influencing consumer WTP premium for green agricultural products, which mainly concerns quality, demographic characteristics, perceptions, and social factors [28,40]. Consumers who are well aware of sustainable labels will pay higher prices for green agricultural products [26,37]. In the case of information asymmetry, it is unlikely that consumers will pay higher prices for sustainable agricultural products if they are unfamiliar with those products, especially when the market is inadequately regulated and the product promotion is missing [41]. It is generally acknowledged that higher income is associated with a higher WTP premium [11,25]. Age is negatively correlated with the decision to consume green agricultural products [7,42], while women have a higher level of WTP premium than men for such products [13,43].

Prices play a role in consumers’ purchase behaviors, and knowledge of price premiums allows for informed marketing decisions by distributors such as e-platform operators [24]. However, previous research on price premiums of sustainable labeling in agricultural products is mostly about WTP premiums, and less attention was paid to the “revealed” price premium. There are very few studies on the price of sustainable agricultural products sold on e-commerce platforms. Therefore, this study examines the price premium of green food-labeled and organic food-labeled agricultural products on e-commerce platforms, which contributes to the literature by exploring the price premium using transaction data. Important insights are gained to promote the sustainable agrifood market.

## 3. Method

The term “hedonic”, originally “hedonistic”, refers to the satisfaction of material desires, which in economics implies the acquisition of utility [44]. Lancaster [45] proposed the concept of an “attribute bundle” based on the heterogeneity of products, which led to the core idea of the theory, maintaining that each product is a blend of attributes, and consumers buy the product for the attribute bundle that affects their utility. That is to say, the product purchase involves a collection of inherent attributes. Moreover, it is the implicit price corresponding to the set attributes that determines the final product price, though the attributes may not be directly observable in the market.

Following Rosen [46], the currently considered empirical relationship between rice prices and product attributes is expressed as follows:(1)$$U=f\left(x,Z_{1},Z_{2},\ldots ,Z_{n}\right),$$
(2)$$P\left(Z\right)=P\left(Z_{1},Z_{2},\ldots ,Z_{n}\right),$$
(3)$$P_{Z_{n}}=U_{Z_{n}}/U_{x}=\partial P/\partial Z_{n},$$

In (1), U is utilities provided by the rice product Z, and $Z_{i}$ is a vector of extrinsic and intrinsic product attributes. In (2), P is the actual transaction price of Z. Consumer choice is based on the utility maximization principle, which involves the choice $\left(Z_{1},Z_{2},\ldots ,Z_{n}\right)\;$ and x items (i.e., other goods) subjected to the consumer budget constraint (y) [33]. As shown (on the left side) of (3), $P_{Z_{n}}$ is the ratio of the utility of one single attribute to the utility of compound product x and objectively reflects the degree of consumer preference for the product. The ratio is referred to as the implicit price of the attribute.

In current studies, the selection of product attributes takes into account both the rice product and the e-commerce platform. A previous study [47] on the price formation of certified rice has shown that the rice variety, brand, and certification positively affect the price, while the rice shape, package, and geographical protection mark lower the rice products’ price premium. In a study of rice purchasing behaviors of urban residents, Cao et al. [48] selected the distribution platform, season when purchasing, variety, and rice origin, quality, brand, package, and shelf life as independent variables. The results showed that more than half of the consumers chose to buy simply packaged rice, and the quality and safety information was of great concern to consumers. In studies of e-commerce platforms [49,50], the delivery time, online security, number of reviews, and proportion of picture reviews have different effects on consumer choice of the e-commerce platform when purchasing agricultural products. Jiang et al. [51] concluded that suitable product display and reputation incentive mechanisms in e-commerce have a significant positive effect in boosting the consumption of green agricultural products.

Based on past studies and currently available data, both intrinsic and extrinsic attributes were chosen in this study. In terms of intrinsic attributes, green food labels and organic food labels were included as proxy variables for the sustainable label. Rice variety and type were chosen to examine the effect of the product quality on the price premium while designating rice as imported was adopted to explore the effect of geographical attributes on price. To account for the extrinsic attributes, the sales platform, market share, package type, and promotion were selected to measure the extent to which external factors contribute to the price premium.

The functions commonly applied in the hedonic price model are linear, log-log, log-linear, and semi-log forms [52]. Nonlinearities are generic features of equilibrium in hedonic models and a fundamental and economically motivated source of identification [53]. The current study selected the log-linear functional form for regression. The empirical hedonic price equation is described as follows:(4)$$\begin{array}{c} Ln\left(P_{i}\right)=\beta_{0}+\beta_{1}\left(Sustainable\;label_{i}\right)+\beta_{2}\left(Sales\;platform{\;}_{i}\right)+\\ \beta_{3}\left(Brands\;with\;high\;market\;share_{i}\right)+\beta_{4}\left(Variety_{i}\right)+\beta_{5}\left(Type_{i}\right)+\\ \beta_{6}\left(Package\;type_{i}\right)+\beta_{7}\left(Import_{i}\right)+\beta_{8}\left(Promotion_{i}\right)+\varepsilon_{i}, \end{array}$$

In (4), $Ln\left(P_{i}\right)$ is the natural logarithm of the transaction price for ith rice (i=1,…,n). The unknown parameters ($\beta_{0},\beta_{1},\beta_{2},\beta_{3},\beta_{4},\beta_{5},\beta_{6},\beta_{7},\beta_{8}$) correspond to eight rice attributes (sustainable label, sales platform, brands with higher market share, variety, type, package type, import, and promotion), and $\varepsilon_{i}$ is the error term that is independently distributed with a mean of zero.

However, the parameters obtained from log-linear estimates cannot be directly interpreted as marginal effects. It is necessary to further calculate their marginal implicit prices (i.e., hedonic prices) using the following formula [54,55]:(5)$$P_{Z_{n}}/P=\{\begin{array}{c} \;\;\beta \;Continous\;variable\\ e^{\beta }-1\;Dummy\;variable \end{array},$$
where β’s are the parameters obtained from the log-linear estimates, P is the transaction price for the baseline product, and $P_{Z_{n}}$ is the hedonic price of attribute $Z_{n}$.

## 4. Data and Descriptive Statistics

### 4.1. Data

Rice is the most important staple food in China, serving as the staple food for more than 60% of the population due to dietary habits [56]. Compared with animal-based products, consumer perception of safety in rice can be remarkably improved by certification logos [57]. Therefore, we take rice as representative empirical data.

In order to confirm the research on the recent market phenomenon of online marketing, this study applies the data from the major domestic food e-commerce platforms. Through the preliminary online survey, we intended to pick several representative online fresh food platforms regarding different age groups of consumers, platform popularity, and platform sales conditions. As a result, seven e-commerce platforms were chosen: COFCO, Freshippo, Tmall, JD, Missfresh, Dmall, and Taobao. Considering that the market data have the advantage that real purchases are used, the data in this paper were first-hand e-commerce data [24]. A total of 200–220 rice products sold on e-commerce platforms were selected at a fixed date every month. This study has a total of 2549 records after 12 months of collection (from November 2019 to October 2020). The data include the price of rice products, green food label, organic food label, date of data obtained, sales platform, brand, rice variety, rice type, country of production, packaging type, and whether promotion/form of promotion. During data washing, 57 outliers were eliminated, referring to the standard of “$\bar{X}\pm 2.5SD$” [58], and finally, there were 2492 pieces of data, with an efficiency of 97.76%.

### 4.2. Descriptive Statistics

The dependent variable in this study is the standard price for a 5 kg bag of rice sold on e-commerce platforms, which is the most common size purchased [59]. Table 2 shows the sample descriptive statistics. The mean of the pooled data is 65.813 CNY/5 kg, the maximum value is 229 CNY/5 kg, the minimum value is 13.190 CNY/5 kg, and the standard deviation is 39.314. Considering that the rice purchased daily is generally homegrown and in simple packaging with significant brand recognition, we also analyzed the price of that kind of rice product. Table 2 (fourth row) shows the average price of 49.925 CNY/5 kg, which is not perceptibly different from a 5 kg bag of rice sold offline, indicating that the price is not substantially different from the rice sales on e-platforms.

To gain an initial understanding of the impact of sustainable labels on the actual premium paid, we compared the average price of different categories of rice products. The average price of ordinary rice products was 58.505 CNY/5 kg, and the average price of rice products with sustainable labels was 94.215 CNY/5 kg. For the several types of sustainable labels, the average price of green food-labeled rice products was 66.741 CNY/5 kg, and the average price of organic-labeled rice products was 129.016 CNY/5 kg.

Table 3 shows the descriptive statistics for extrinsic and intrinsic attributes of rice on e-commerce platforms. The current study focuses on rice products with a sustainable label, including two classifications: the green food label and the organic food label. There are a total of 510 products with a sustainable label, of which 285 and 225, respectively, are green food-labeled rice products and organic food-labeled rice products. The two types account for 11.4% and 9.0% of the total, respectively. In terms of sales platforms, the proportion of rice product sales from each e-commerce platform is Dmall (24.9%), Freshippo (15.4%), JD (12.8%), Missfresh (5.3%), Taobao (15.5%), Tmall (14.5%), and COFCO (11.7%). Brands with high market shares include FuLinMen, QiHeYuan, ChaiHuoDaYuan, ShiYueDaoTian, JinLongYu, and COFCO ChuCui, respectively, accounting for 35.2% out of 2492 observations. Among varieties, the top four varieties are DaoHuaXiang rice (27.3%), long-grain rice (20.4%), Komachi rice (6.0%), and Jasmine rice (5.1%). According to the classification of paddy, rice can be divided into two categories: japonica and indica, with the former accounting for 88.8%, mostly grown in northern areas, and the latter accounting for 11.2%, mostly grown in southern areas. As for the package type, 96.3% of the rice products have simple packaging, such as the plastic sealed vacuum bag. A few (3.7%) of the rice products are finely packaged, for example, in gift boxes. Regarding the origin, the vast majority of rice products sold on e-commerce platforms were grown in China (92.4%). Only a small percentage originates from other countries, including Thailand, Cambodia, or Japan. Additionally, from the perspective of marketing, more than half of the rice products (51.9%) are on sale.

## 5. Results and Discussion

Rice is a multi-attribute product, with each product representing a different bundle of attributes noted by the sustainable label, origin, brand, variety, type, package, marketing strategy, and price. The estimation results of two pooled data models and two label segment models of the hedonic price equations are shown in Table 4.

Organic food has a stricter standard than green food according to China’s food quality standards. The model in the second column of Table 4 shows estimation results of rice products with green food labels and organic food labels as having a sustainable label. Those results allow examination of whether there is a price premium for rice products with sustainable labels and what is the magnitude of the premium. Two label segment models are intended to further examine the magnitude of premiums for different sustainable labels (i.e., green food label and organic food label). Another pooled data model listed in the third column of Table 4 classifies rice products into green food, organic food, and ordinary food by label, and the robustness of the estimated hedonic price model is established by comparing test results with other models.

As shown in Table 4, the regression coefficients and their significance for the sustainable label (i.e., green food label and organic food label) of the model (Table 4, 3rd column) are not obviously different from the regression coefficients and their statistical significance level of the two label segment models. Meanwhile, the regression coefficients and their significance across models are also quite similar. Thus, the hedonic price model is robust, and the variables are appropriately selected.

The pooled data model in Table 5 provides evidence that there is a significant price premium indeed for the rice products with a sustainable label, and at the significance level of 1%, the premium for the rice products with a sustainable label compared to ordinary rice products is +47.55% ceteris paribus. When further reviewing the results for different label types, the premium paid for rice products with the green food label is +19.84%. However, previous studies [9,60] calculated an average consumer willingness to pay for green rice to be 25–50%, higher than the result of this study, showing that there is a large efficiency loss conversion of consumer willingness to pay into the price premium. The price premium for rice products with the organic label is 91.55%, which is considerably higher than the outcomes of previous studies [61] on the willingness to pay for organic food. The result indicates a higher value of organic agricultural products to consumers. Moreover, different standards of sustainable certification can generate differential price premiums.

Taking Dmall as the baseline, Freshippo, Taobao, and COFCO have significant positive premiums of 31.52%, 26.62%, and 15.37%, respectively, while JD and Tmall have price discounts of 17.47% and 17.55%, respectively, and Missfresh does not differ from the omitted platform. The results show that the premium amount affected by professional fresh food e-commerce platforms such as COFCO, Dmall, and Freshippo is higher than non-professional e-commerce platforms such as JD and Tmall. It appears that consumers pay a premium for professional fresh food e-commerce platforms and non-professional e-commerce platforms with product quality assurance [50].

Rice variety generates a price premium for DaoHuaXiang rice (+48.44%) and a price discount for Komachi rice (−10.95%) in comparison to other rice varieties. The reason may lie in the fact that DaoHuaXiang rice is mainly produced in Northeast China, where the fertile black land is more suitable for rice production and benefits from this reputation.

The price premium for finely packaged rice products, such as rice in gift boxes, is 65.86%. The result proves that packaging, as an additional element, reflects the value of the agricultural products, potentially contributing to the actual price premium. Wu [47] showed that packaging was not of high utility to consumers when rice was packed in ordinary woven bags and vacuum packaged. The current study differs from Wu’s [47] in its definition of package type and distinguishes between simple and fine packages.

Finally, there is a significant price premium for imported rice products over rice products from China (Table 4). The magnitude of the premium is 47.99% assuming all other attributes are the same. The premium may be influenced by a preference for geographical factors or transaction costs such as taxes. Moreover, the price premium for promotional rice products is −15.13%.

## 6. Conclusions and Recommendations

### 6.1. Conclusions

In response to the insufficient exploitation of the market potential regarding sustainable agricultural products, this study considers rice sold on e-commerce platforms and specifies a hedonic price model to explore the impact of sustainable labels on the price of a common agricultural product, rice.

Results allow for drawing the following conclusions. First, there is a significant price premium (47.55%) for rice products with a sustainable label, and different standards of certification can generate variations in the price premium. Specifically, the price premium for rice with an organic food label is higher than that of rice with a green food label. Second, the actual price premium for the green food-labeled rice products obtained in this study is lower than the willingness to pay calculated by previous studies. The discrepancy indicates that there is a large efficiency loss in the conversion of consumer willingness to pay into price premiums. The result suggests careful consideration of the method used to obtain the possible price premium in making management decisions, including the expansion of production and marketing. Third, in addition to the sustainable label, factors that significantly affect rice price premium include e-commerce platform, package type, rice variety, and origin. It suggests that the price of agricultural products is determined by a variety of extrinsic and intrinsic attributes that require consideration prior to choosing e-commerce platforms for product distribution.

### 6.2. Recommendations

Based on the main conclusion, this study provides insights on how to further develop green food and organic food labeling and improve the market for sustainable food and agricultural products.

Agribusinesses, cooperatives, and individual farmers should be encouraged to consider sustainable agricultural production and certifications. Food consumption markets, especially organic food, green food, and other high-grade food products, offer much potential and have been underexplored. This study suggests that both green food and organic food-labeled products have a significant price premium. Therefore, sustainable agricultural production is not only environmentally friendly but also provides economic incentives for agricultural producers. At the early stage of transitioning from conventional agricultural production to sustainable production, the price premium may not fully cover the cost increase among small-holder farmers. However, reasonable subsidies can encourage producers to join the sustainable agricultural industry, adhere to industry standards, and supply high-quality agricultural products while earning price premiums.

Making consumers aware of and enhancing trust in high-quality, sustainable agricultural products requires sustained communication and quality verification. The expansion of the emerging e-commerce platforms is desirable for sustainable agricultural products to account for a larger share of purchased foods. Online shopping platforms can highlight the differences between agricultural products and have a certain promotional effect on the development of high-quality agricultural products. Participation in online sales allows for obtaining information more accurately, describing the targeted consumers. By reducing business costs and improving operational efficiency while further enhancing brand recognition, online sales improve the economic performance of local producers and promote the development of a sustainable agricultural market.

### 6.3. Limitations and Future Studies

This study examined the price premium of sustainable labeling (green and organic food labeling) using rice in the e-commerce platform as an empirical case. However, due to the lack of data, this study used only one year of data as the sample. If a broader range of data could be applied, more valuable studies might be investigated, such as capturing the changes and trends of the price premium in the long run. Moreover, a further comparative analysis that includes offline and online channels in the same framework would provide a more holistic understanding of consumer behavior toward sustainable labels.

## Figures and Tables

**Table 1 foods-11-03781-t001:** Description of sustainable labels.

Certification	Green Food	Organic Food
Label	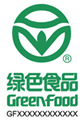	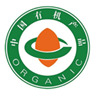	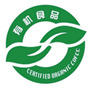
Certificate authority	China Green Food Development Center, Ministry of Agriculture and Rural Affairs of the People’s Republic of China	China Organic Food Certification Center, Ministry of Agriculture and Rural Affairs of the People’s Republic of China; Organic Food Development Centre, Ministry of Ecology and Environment of the People’s Republic of China
Validity of certificated label	3 years	1 year
Operation year [8]	1990	1994
Main differences [8,10]	Controlled and limited use of synthetic fertilizer, pesticides, growth regulators, livestock and poultry feed additives, and gene engineering technology. (A-class green food)	No use of synthetic fertilizers, pesticides, growth regulators, livestock and poultry feed additives, and gene engineering technology. (AA-class green food, suspended after June 2008)

**Table 2 foods-11-03781-t002:** Descriptive statistics for prices of rice on e-commerce platforms.

Characteristic	Mean	Std Dev	Max	Min	Count
Price (CNY/5 kg) of full sample	65.813	39.314	229.000	13.190	2492
Price (CNY/5 kg) of ordinary rice	58.505	32.555	229.000	13.190	1982
Price (CNY/5 kg) of ordinary rice, homegrown, and simply packaged with significant brand recognition	49.925	19.745	133.333	24.875	721

**Table 3 foods-11-03781-t003:** Descriptive statistics for the extrinsic and intrinsic attributes of rice sold on e-commerce platforms.

Variable	Distinct	Value	Frequency	Proportion
Sustainable label	3	Ordinary	1982	0.795
		Green food label	285	0.114
		Organic food label	225	0.090
Sales platform	7	Dmall	620	0.249
		Freshippo	383	0.154
		JD	318	0.128
		Missfresh	132	0.053
		Taobao	387	0.155
		Tmall	361	0.145
		COFCO	291	0.117
Brands with higher market share	2	0 = No	1615	0.648
		1 = Yes	877	0.352
Variety	5	DaoHuaXiang rice	681	0.273
		Jasmine rice	126	0.051
		Komachi rice	149	0.060
		Long-grain rice	509	0.204
		Other	1027	0.412
Type	2	0 = Indica rice	280	0.112
		1 = Japonica rice	2212	0.888
Package type	2	0 = Simple package	2400	0.963
		1 = Fine package	92	0.037
Import	2	0 = No	2302	0.924
		1 = Yes	190	0.076
Promotion	2	0 = No	1199	0.481
		1 = Yes	1293	0.519

**Table 4 foods-11-03781-t004:** Robustness tests.

Y=Ln(P)	Pooled		Label Segments	
	Green Food and Organic Food Label	Green Food/Organic Food Label	Green Food Label	Organic Food Label
(Intercept)	3.858 *** (0.053)	3.847 *** (0.052)	3.819 *** (0.052)	3.835 *** (0.056)
Sustainable label	0.389 *** (0.024)	0.181 *** (0.026)	0.180 *** (0.026)	0.651 *** (0.032)
	-	0.650 *** (0.032)	-	-
Other attributes	YES	YES	YES	YES
Num of observations	2492	2492	2267	2207
$R^{2}$	0.484	0.519	0.442	0.540

Note: Months have been controlled. Robust standard errors in parentheses correcting autocorrelation and heteroskedasticity. *** *p* < 0.01.

**Table 5 foods-11-03781-t005:** Estimation results of sustainable labels on rice prices on an e-commerce platform.

Y=Ln(P)	Pooled (Green Food and Organic Food Label)	
	β	Percentage of Price Premium
(Intercept)	3.858 *** (0.053)	-
Sustainable label	0.389 *** (0.024)	+47.55%
Sales platform (Baseline = Dmall)		
Freshippo	0.274 *** (0.027)	+31.52%
JD	−0.192 *** (0.028)	−17.47%
Missfresh	0.000 (0.025)	+0.00%
Taobao	0.236 *** (0.030)	+26.62%
Tmall	−0.193 *** (0.027)	−17.55%
COFCO	0.143 *** (0.032)	+15.37%
Brands with higher market share	−0.016 (0.019)	−1.59%
Variety (Baseline = Other)		
DaoHuaXiang rice	0.395 *** (0.021)	+48.44%
Jasmine rice	0.054 (0.060)	+5.55%
Komachi rice	−0.116 *** (0.032)	−10.95%
Long-grain rice	−0.010 (0.021)	−1.00%
Type	0.049 (0.042)	+5.02%
Package type	0.506 *** (0.040)	+65.86%
Import	0.392 *** (0.055)	+47.99%
Promotion	−0.164 *** (0.018)	−15.13%
Num of observations	2492	
$R^{2}$	0.484	

Note: Type: 1—Japonica rice; 0—Indica rice. Package type: 1—Fine package; 0—Simple package. Months have been controlled. Robust standard errors in parentheses correcting autocorrelation and heteroskedasticity. *** *p* < 0.01.

## Data Availability

The data that support the findings of this study are available from the corresponding authors upon reasonable request.

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
