# Peer review of "Impact of Food Sustainability Labels on the Price of Rice in Online Sales"

_foods, 2022, doi:10.3390/foods11233781_

Round 1

Reviewer 1 Report

The proposed manuscript aims to analyse the impact of the sustainable label on the price of rice sell on e-commerce platforms. The paper covers an interesting topic and the theme is very timely for the sector. 

The paper is well written and easy to read.

Some suggestions seem to be necessary to improve the paper. A minor revision is required. 

Line 168, change “marketing” to “marketing”

Line 226, add a space before the “note”

Conclusions must be improved. 

Line 410, send to next line the reference "Wongprawmas etl al, 2017"

Author Response

Dear Reviewer, 

Thank you very much for taking time to thoroughly review our revised manuscript. Please find our detailed response letter as attached. 

Best,

Authors

Reviewer 2 Report

The writing and arguments thorughout the paper must be substantially improved (for examples, see specific comments below). 

The literature review should contain a review of hedonic price studies of which there are many. The current litt review containing very litle hedonic literature is hardly relevant for the research design and methods applied.

Much more information about the eco-labels studied must be provided. Certification process, independent third-party certification or not, labelling types, logos, etc.

More information about the data must be provided.

 Specific comments:

Line 31: What is "biodynamic agriculture"? And please provide some reference(s) for the claim you make regarding the rise of organic farming in the 1920s.

Line 32: This sentence is very short and needs a lot more elaboration as well as references.

Line 34: what is "green" food? What types of environmental issues are covered in the standard? Is the standard followed by some labeling scheme? What does the label look like and what information is included? Is a third-party involved in certification?

Line 35-36: How is AA-class green food equivalent to organic food?

Line 37-38: What about "green" issues? Please provide references to back your claims.

Line 41: What is the exchange rate for Euros or USD?

Line 44: What is green agricultural products? Are these certified products?

Line 43-44: Provide reference to support claim.

Line 45-46: Provide reference to support claim. There is a substantial literature on price premiums for eco-labels. For seafood products, check authors such as Asche, F., Roheim, C., Bronnmann, J. and Sogn-Grundvåg, G.

Line 47-48: That price premiums at the retail level is transmitted to producers is not clear from the literature. 

Line 49: What is "agroecological environment"?

Lines 49-50: Consumer willingness to pay for what? Eco-labels?

Lines 52-54: Provide reference to support claim.

Lines 56-57: Are there any reasons why e-commerce should be better suited for eco-labeled products? 

Line 57: What do you mean by "Information Age"?

Line 59-60: This is a complete misunderstanding. When a product is perceived to be environmentally friendly, it enhances consumers' perceived value of the product, which leads to a price premium.

Line 62: I could not find the Xuan et al 2004 paper when searching google schoolar and the internett. And the journal "Consumer Economics" does not seem to exist. And the Lu et al (2017) reference is not in the reference list.

Lines 66-67: influencing price of what? Eco-labeled products? There is a substantial hedonic literature on eco-labeled seafood (see the authors mentioned above). And these focus on product attributes such as quality. 

Lines 85-86: This is a strong claim given that you have not reviewed the hedonic literature focusing on price premiums for eco-labels.

Lines: 88-89: Please provide more information about your approach. Will you study consumers and/or other links in the supply chain? If you study consumers you cannot say much about production etc because any price premiums is unlikely to be transmitted to producers. See e.g. papers by Stemle et al (Fisheries Research) and Blomquist et al (European Review of Agricultural Economics).  

Line 116: The Wu reference is not possible to find on the internet.

Lines 141-145: Why do you report results here?

Line 168: Please provide more detail on the data.

Line 169: "Online survey and offline visits" What is this?

Lines 172-175: What is a "one sheet per month"? Please provide more information about the nature of the data.

Line 174: "2,549 pieces of data" What is this? Is it transactions or does each piece contain many transactions? This is very unclear!

179: 980 is an extremly high price compared to the mean. How is this possible for rice?

Table 2 contains too little information to be a table. Use the prices in the text please.

Line 226-227: You say in the introduction that they are equivalent. Please clarify.

Author Response

(The authors gave the same response as above.)

Reviewer 3 Report

This paper is interesting as it examines, according to the authors, rice on Chinese e-commerce platforms (as an entry point) and constructs a hedonic price model to explore the impact of the sustainable label on the price of agricultural products. The paper is seeming to be well-organized and structured and uses the appropriate methodology, however, the topic has a clear national, Chinese, interest. The novelty of the topic is based on data from China, but the research seems to not go in-depth. Some improvements in the paper could nevertheless be useful to show interest.

Comments

1.      In the Abstract the purpose of the paper does not present clearly. The Introduction section also missing at the end of the section the objective of the research (research questions or hypotheses also). In the end, a brief structure of the paper is needed.

2.      Generally, the introduction section is poor (and the literature review that follows does not help also). The scientific object, of the paper, is so wide and the international literature review so rich that makes the paper loses its novelty from the other hand it is necessary for the reader to have a full and clear image of the literature background.

3.      In a so brief Literature review (section 2), I found repetitions (look at the first paragraph). Please avoid repetitions they add nothing.

4.      Why did you choose as the dependent variable the standard price of a 5 kg bag of rice sold on e-commerce platforms? Why not adopt the standard price of 1kg? Please documented your choice.

5.      Generally, section 4 missing the documentation through secondary data of the research design, for example, why the rice? why the e-commerce platforms? why this dependent variable? etc.

6.      The conclusions subsection is a simple recap of everything the authors mention in the previous section. There is no generalization of the results as a conclusion, but the recommendations subsection is interesting.

7.      There are no research limitations and future research.

8.      Authors do not present the contribution of the research and its originality.

9.      References are limited and need to be enriched. Important references are missing.

Author Response

Dear Reviewer,

Thank you very much for taking time to thoroughly review our revised manuscript. Please see the attachment for our detailed response. 

Best,

Authors

Round 2

Reviewer 3 Report

No comments